# The Effects of Social Media Use on the Health of Older Adults: An Empirical Analysis Based on 2017 Chinese General Social Survey

**DOI:** 10.3390/healthcare9091143

**Published:** 2021-09-01

**Authors:** Liping Fu, Yu Xie

**Affiliations:** 1School of Public Administration, College of Management and Economics, Tianjin University, Tianjin 300072, China; lpf3688@126.com; 2Center for Social Science Survey and Data, Tianjin University, Tianjin 300072, China

**Keywords:** social media, older adults, public health, active aging

## Abstract

An aging population and social informatization are currently the two main social phenomena affecting China. Under their influences, the real-life experiences of older adults are becoming more and more closely connected to the online world, and the influences of the Internet on healthy aging cannot be ignored. This work aimed to study whether Internet use had an effect on the physical and mental health of older adults, whether the effect was positive or negative, and whether its influence on physical and mental health was heterogeneous. In this study, data from the 2017 Chinese General Social Survey (CGSS) was used to analyze the effects of social media use on the physical and mental health of older adults. The results indicated that there was a significant positive relationship between social media use and the health of older adults. The correlation between social media use and mental health of older adults was more significant than physical health. These results could help us further study the effects of Internet use on the health of older adults.

## 1. Introduction

According to the Seventh National Census of China, as of 2020, there were 260 million people aged 60 and above in China, accounting for 18.7% of the total population [1]; this represents an increase of 5.44% within ten years. From the perspective of active aging, the health of older adults has attracted serious concern, as health determines the quality of life for older adults. For the country as a whole, health is the prerequisite and basis for active aging [2].

China is also in the process of high-speed informatization. With the popularization of the Internet and the development of smartphones and other devices, people’s daily lives are more and more closely connected with the Internet. According to the 47th China Statistical Report on Internet Development in China, as of December 2020, there were 989 million netizens in China, among whom about 110 million were over 60 years old, accounting for 11.2% of the total [3]. Elderly netizens have played important roles in the Internet world, and their Internet use will have an important effect on their lives [4,5,6].

Existing studies claim that Internet use can influence the health of older adults. For example, most studies have indicated that Internet use can significantly improve the mental health of older adults [7], although some have shown that Internet use has a negative or no significant effect [8,9,10]. Internet use by older adults can relieve their loneliness and improve their mental health [11,12,13]. Moreover, it can improve their subjective well-being and life satisfaction [14,15,16]. Older adults who use the Internet have lower rates of mental illnesses such as depression [17,18,19]. There are relatively fewer studies on physical health. It is generally believed that older adults can gain health knowledge and access services on the Internet that can enable them to improve their health condition [2,20], and that they can reduce the morbidity of chronic diseases by sharing information with others on the Internet [21,22]. In addition, Internet use can improve the self-rated health status of older adults [23,24].

Currently, studies on the impact of social media use on health have mainly focused on subjective well-being. Research has indicated that Facebook can serve as an effective tool for resisting isolation and promoting socialization [12]. Social media such as QQ and Weibo can also affect the subjective well-being of users, with a positive correlation shown [25,26]. Widespread use of social media can improve overall happiness [27]. Studies of older adults have indicated that their social activities on the Internet can improve their subjective well-being [28,29,30] and that the use of WeChat by the older adults in cities has had an auxiliary effect on improving subjective well-being [31].

How Internet use affects the health of older adults is a current research hotspot and priority. Importantly, the Internet can improve the mental health of older adults by alleviating loneliness. The Internet can help older adults overcome the barriers of time and distance and increase the frequency of social interaction [18,32,33]. The Internet can also help older adults increase their social capital and maintain social contact [13], each of which can help them combat loneliness. Improving happiness is also an important factor in the Internet’s promotion of mental health in older adults. Self-expression on social media [26], the use of the Internet to promote self-cognition [29], and learning, social, and entertainment activities [30,34] can all improve the happiness of older adults. Internet use promotes the health of older adults by strengthening social support. Short video applications can expand the social network of older adults [35]; the network community can provide social support, promote the self-protection of older adults and provide an opportunity for self-discovery and growth [36]; and virtual communities can provide a sense of belonging for older adults to assist in the reimplementation of social embeddedness [37] and provide social participation opportunities for older adults [38]. The health information provided on the Internet can improve people’s diet and exercise methods [18] and assist users in pain self-management [39], thus improving the life and health of users. The role of the Internet in promoting physical health is mainly realized through interaction, information sharing, discussion, and technical support. The Internet provides a platform for patients and medical staff to keep in touch and actively interact [40,41]. The use of discussion boards on websites can enhance users’ learning experiences and influence their health behaviors [42]. The Internet can also provide technical support to caregivers for older adults [43]. In addition, active participation in Internet use can slow cognitive decline and improve health [44].

Existing studies have extensively examined the impact of Internet use on the health of older adults, but they have some shortcomings. First, there have been limited studies on specific Internet use behaviors. Most studies have focused on discussing the effects of frequency of the Internet use on older adults, and there has been a lack of detailed descriptions of older adults’ specific online behaviors. Second, there has been insufficient analysis of the heterogeneity of the effects of Internet use on the health of older adults. Existing studies have mostly focused on the mental health of older adults, and there have been rare studies discussing the effect of Internet use on physical health. Third, there has been a lack of studies using representative nationwide data. Studies on the use of social media have mostly used sample sizes or local data from a single province or city.

Based on the above knowledge gap, our research scope was: first, to examine whether social media use had an effect on the physical and mental health of older adults and, if so, to examine whether this effect was positive or negative; second, to analyze whether the effects of social media use on the physical and mental health of older adults showed heterogeneity; and third, combined with existing research, to explore the mechanisms of the effects of social media use on the physical and mental health of older adults.

## 2. Methods

### 2.1. Data Sources

The data used in this study are from the 2017 Chinese General Social Survey (CGSS) [45]. This survey is China’s first national, comprehensive, and continuous large-scale social survey project, hosted by the China Survey and Data Center at Renmin University of China. The purpose of the survey is to summarize long-term trends of social change by regularly and systematically collecting data on the Chinese people and various aspects of Chinese society. In 2017, the CGSS was conducted using a questionnaire survey, and a total of 12,582 valid samples were completed. The data collected by this survey cover 125 counties in 28 provinces, autonomous regions, and municipalities in China, involving multiple levels of society, communities, families, and individuals, and it was both systematic and comprehensive. In 2017, the CGSS questionnaire was composed of three modules, namely the CORE module A, the social network and network society module C, and the family questionnaire module D. The published data contain 738 variables. Module A comprises 12 parts including social and demographic attributes, health, migration, lifestyle, class identity, and family. Module C includes two parts: social network and network society.

The China Survey and Data Center at Renmin University of China is a comprehensive research institute directly under Renmin University of China, and it has joined the International Community Survey Cooperation Organization on behalf of the country. The capacity of the center ensures that its research process is scientific, systematic, and comprehensive. In addition, the CGSS data set, the first social science survey data database in the country, is part of the China National Survey Database, established by the China National Survey and Data Center under the commission of the National Natural Science Foundation of China. Therefore, the CGSS data set is reliable. Both modules A and C of the CGSS2017 contain questions about residents’ Internet use and consist of rare data that are nationally representative of present-day individual Internet use in China. It is also aimed at studying the effects, including health effects, of Internet use on the population. The purpose of this paper is to study the relationship between the use of social media and the health of the older adults. Module A of CGSS2017 contains data related to the health of the elderly, and module C contains data on the use of social media by older adults. In addition, module A also contains data on the socio-demographic attributes, class identities, and family situations of the older adults. Therefore, these data are compatible with the research questions in this paper. The 2017 data are the most recent that are available from the agency.

The research objects of this study were older adults over 60 years old. According to the issues studied, with missing and invalid variables removed, a total of 1278 valid samples were obtained.

### 2.2. Variable Selection

CGSS2017 obtains data through a household questionnaire survey, and the variables selected in this paper were measured by selecting appropriate questions from the CGSS2017 questionnaire [45]. The selected questions were as follows: A16 and A17 in the health section of module A; A2 and A3 in the social population attribute section; A43a in the class identity section; A62 and A63 in the family section; and C27 in the social network section of module C. The details of each question, option settings, and assignment are described below.

This paper analyzed the health of older adults from the two aspects of physical and mental health [46]. Question A16 was selected to measure the explained variable of physical health, i.e., “How often has your work or other daily activities been affected by health problems in the past four weeks?” The respondents’ answers were coded as 1 = always, 2 = often, 3 = sometimes, 4 = rarely, 5 = never. Question A17 was selected for the explained variable of mental health, i.e., “How often did you feel depressed in the past four weeks?”. The respondents’ answers were coded as 1 = always, 2 = often, 3 = sometimes, 4 = rarely, 5 = never. The lower the assigned values for the respondents’ answers, the worse the indicated corresponding health status. This study concerned the effect of social media use on the health of older adults, so question C27 was selected for measurement of the explanatory variable of social media use, i.e., “Now think about your contacts with all your family members and close friends; how many of them are conducted through WeChat, mobile phones or other network communication devices?” According to the respondents’ answers, values were assigned as 1 = all or almost all, 2 = most, 3 = about half, 4 = some, 5 = none or almost none, 6 = I have not used any of these devices. The lower the assigned value for the respondents’ answers, the more frequent the use of social media was indicated.

Other variables were set as follows according to the two dimensions of personal characteristics and social characteristics: “gender”, “age”, and “household registration” were selected as personal characteristics, and “social status”, “family income”, and “family size” were selected as social characteristics. Question A2 was selected for gender with an assigned value of 1 = male and 2 = female, i.e., “Gender”. Question A3 was selected for age, i.e., “What is the date of your birth?”, and converted to the age of the respondent according to the year. For example, if a respondent was born in 1957, the age was recorded as 60 years old. The variable “is urban” in CGSS2017 data set was selected for household registration, with assigned values of 1 = urban and 2 = rural. Question A43a was selected for measuring social status, i.e., “On the whole, in the current society, which level of society you are in?” Respondents rated their social status from 1–10, with 10 representing the top level, and 1 representing the lowest level. Question A62 was selected for measuring household income, i.e., “What is the total income of your family in 2016?”, with “thousand RMB” as the unit. Personal income was not selected as a variable to measure the economic level because in China sons and daughters usually support their older parents [47], and thus, the economic level of older adults often depends on income sources other than their own. Question A63 was selected for measuring family size, i.e., “How many people currently live together in your family?”, with the respondent himself or herself included in the number of members calculated.

### 2.3. Data Analysis

SPSS 24.0 software was used to analyze all data in this study. First, descriptive statistics were created for all variables to understand their basics. Next, it can be seen in Section 2.2 that the physical and mental health variables of older adults, rated from 1 to 5, represented the continuous improvement of their health status and could be regarded as ordered variables. Therefore, multivariate ordered logistic regression was used to further study the influence of social media use on the health status of older adults. The physical health model (Model 1) is as follows:(1)$$\begin{array}{c} \mathrm{Logit}\left[P\left(Health_{p}=1\right)\right]=\ln\left(\frac{\pi_{1}}{1-\pi_{1}}\right)=\beta_{0}^{1}+\beta_{s}X_{s}+\overset{6}{\underset{i}{\sum }}\beta_{i}x_{i}\\ \mathrm{Logit}\left[P\left(Health_{p}=2\right)\right]=\ln\left(\frac{\pi_{1}+\pi_{2}}{1-\pi_{1}-\pi_{2}}\right)=\beta_{0}^{2}+\beta_{s}X_{s}+\overset{6}{\underset{i}{\sum }}\beta_{i}x_{i}\\ \mathrm{Logit}\left[P\left(Health_{p}=3\right)\right]=\ln\left(\frac{\pi_{1}+\pi_{2}+\pi_{3}}{1-\pi_{1}-\pi_{2}-\pi_{3}}\right)=\beta_{0}^{3}+\beta_{s}X_{s}+\overset{6}{\underset{i}{\sum }}\beta_{i}x_{i}\\ \mathrm{Logit}\left[P\left(Health_{p}=4\right)\right]=\ln\left(\frac{\pi_{1}+\pi_{2}+\cdots +\pi_{4}}{1-\pi_{1}-\pi_{2}-\cdots -\pi_{4}}\right)=\beta_{0}^{4}+\beta_{s}X_{s}+\overset{6}{\underset{i}{\sum }}\beta_{i}x_{i} \end{array}$$

In Formula (1), $Health_{p}$ represents the physical health of older adults, category 5 (never) serves as the reference class, and $\pi_{i}$ represents the probability of each category of physical health. $X_{s}\;$represents the older adults’ social media use; $x_{i}$ represents the control variables that influences the health of older adults; 1–6 respectively correspond to “gender”, “age”, “household registration”, “social status”, “family income”, and “family size”; and $\beta_{0}^{i}$, $\beta_{s}$, $\beta_{i}$ represent the parameters to be estimated. Similarly, the mental health model (Model 2) can be established as follows:(2)$$\begin{array}{c} \mathrm{Logit}\left[P\left(Health_{m}=1\right)\right]=\ln\left(\frac{\pi_{1}}{1-\pi_{1}}\right)=\gamma_{0}^{1}+\gamma_{s}X_{s}+\overset{6}{\underset{i}{\sum }}\gamma_{i}x_{i}\\ \mathrm{Logit}\left[P\left(Health_{m}=2\right)\right]=\ln\left(\frac{\pi_{1}+\pi_{2}}{1-\pi_{1}-\pi_{2}}\right)=\gamma_{0}^{2}+\gamma_{s}X_{s}+\overset{6}{\underset{i}{\sum }}\gamma_{i}x_{i}\\ \mathrm{Logit}\left[P\left(Health_{m}=3\right)\right]=\ln\left(\frac{\pi_{1}+\pi_{2}+\pi_{3}}{1-\pi_{1}-\pi_{2}-\pi_{3}}\right)=\gamma_{0}^{3}+\gamma_{s}X_{s}+\overset{6}{\underset{i}{\sum }}\gamma_{i}x_{i}\\ \mathrm{Logit}\left[P\left(Health_{m}=4\right)\right]=\ln\left(\frac{\pi_{1}+\pi_{2}+\cdots +\pi_{4}}{1-\pi_{1}-\pi_{2}-\cdots -\pi_{4}}\right)=\gamma_{0}^{4}+\gamma_{s}X_{s}+\overset{6}{\underset{i}{\sum }}\gamma_{i}x_{i}\; \end{array}$$

In Formula (2), $Health_{m}$ represents mental health, category 5 (never) serves as the reference class, and $\pi_{i}$ represents the probability of each category. $X_{s}\;$represents use of social media; $x_{i}$ represents the control variable that influences health; 1–6 respectively correspond to “gender”, “age”, “household registration”, “social status”, “family income”, and “family size”; and $\gamma_{0}^{i}$, $\gamma_{s}$, $\gamma_{i}$ represent the parameters to be estimated.

## 3. Results

Descriptive statistics for all variables are shown in Table 1.

As shown in Table 1, males and females in the sample were equal, both accounting for 50%. The number of urban older adults was slightly higher than rural, accounting for 59%. The average age of the respondents was 69.3 years old, the average annual household income was 51.22 thousand RMB, and the average number of household family members was 2.46. The average social status was 4.08, which is in the middle. Table 2 shows the descriptive statistical results of the explained variables and the core explanatory variables in more detail.

As shown in Table 2, the mean physical health of older adults who used social media extensively (all or almost all, most, about half, some) was 3.57, which was higher than the overall mean (3.48) as well as the mean (3.28) of older adults who did not use social media or who used it only moderately (none or almost none, I have not used any of these devices). The mean mental health of older adults who used social media extensively was 3.80, which was higher than both the overall mean (3.75) and the mean (3.62) of the others. The older adults who used social media extensively were in better health conditions than those who did not use social media or who used it only moderately. The results of the multivariate ordered logistic regression are shown in Table 3.

Model 1 in Table 3 represents physical health. As shown by the model regression results, there was a significant relationship between social media use and the physical health of older adults. Older adults who used social media more often were in better physical health. Model 2 represents mental health. As shown in the results of the regression model, there was a significant correlation between social media use and the mental health of older adults. Older adults who used social media more often were in better mental health. The absolute value of the regression coefficient for social media use in model 2 is larger than that in model 1, and the significance level is higher. This means that, compared with the physical health, there was a more significant correlation between social media use and mental health of older adults.

In terms of personal characteristics, gender, and household registration had a significant effect on both the physical and mental health of older adults. The probability ratio of males having better physical health than females was 1.47 times (exp(0.386)), and the probability of better mental health was 1.43 times (exp(0.355)). The probability ratio of urban older adults being physically healthier than rural ones was 2.06 times (exp(0.723)), and the probability of better mental health was 1.60 times (exp(0.471)). Although the physical health of older adults decreases with age, age had no significant effect on the mental health of older adults. In terms of social characteristics, it can be concluded from models that family income and social status have a significant positive effect on both the physical and mental health of older adults. The higher the family income of older adults, the better their health status. The higher the social status of older adults, the better their health status. Family size has no significant effect on either the physical or mental health of older adults.

## 4. Discussion

The results of the data analysis showed that social media use contributed to the health of older adults. However, some studies have found that Internet use has either no effect or hinders health in older adults [8,9,10]. Older adults use the Internet for different purposes, and different forms of participation may have different impacts on their health. Previous studies that only looked at whether and how often older adults used the Internet did not find that specific Internet behaviors impacted the health of older adults. Therefore, studying the influences of different Internet use behaviors on older adults is meaningful, as it can tell us what kind of Internet activities older adults should be encouraged to engage in. We also found that social media use had different effects on physical and mental health in older adults, with a more significant correlation with mental health. This difference, which has been overlooked in previous studies, can help us better understand how social media affects the health of older adults. Social media use affects a number of health-related factors, but it may affect them in different ways, and understanding these differences can help us determine our own strategies and priorities in terms of influencing the health of older adults through social media. In addition, this paper is different from existing studies using local data in that it uses a large national sample to verify the relationship between social media use and the health of older adults, providing a reference for further research. Based on the above results and previous studies, this paper attempts to analyze how social media use affects the health of older adults and why its correlation with mental health is more significant.

Social media has increased levels of health management for older adults. The self-management levels of older adults are low [48]. The health literacy of older adults may be insufficient, as the health knowledge and health literacy of older people are relatively poor compared to those of other groups. In addition, memory loss and other factors reduce the self-management levels of older adults. Therefore, increasing health knowledge and improving management capacity can improve the health of older adults [7]. Social media has made it more convenient to obtain health knowledge, which has been an important means of older adults improving their health literacy and maintaining their health [49]. According to existing research, health is a major concern for elderly Internet users [20]. When older adults use social media, they can obtain and share relevant content through private chats or groups [41,42]. Because they are restricted by various factors, it is difficult for them to gain the required health knowledge offline in a comprehensive, concentrated and convenient manner. In addition to social attributes, social media also contain information dissemination functions. For example, via WeChat, older adults can receive health knowledge in private chats and group chats and can also browse health information on interfaces such as Moments, Search, and Official Accounts, significantly lowering the difficulty of obtaining health information. In addition, older adults can access information about health resources through the Internet [34]. The Chinese government has taken many measures to improve the health of older adults [50], and a large number of non-governmental organizations have also launched special services [51], such as free regular physical examinations and health consultations as well as the distribution of medical supplies. Older adults can learn how to access these resources and services by sharing them with friends and family members. The use of social media indirectly enhances the health management abilities of older adults. The health managers of older adults include family members and friends apart from themselves [48]. Social media relies on Internet carriers, which are not limited by time and space. The communication of users is highly real-time, and other health managers can communicate with older adults in a timely fashion, which improves the efficiency of health management [22,40].

It is important to note that there is not necessarily a link between information and behavior, which means that older adults may not follow up on health information even if they receive them. Moreover, older adults may not fully understand the health information they obtain [44]. This can explain, to a certain extent, why social media use is less strongly associated with physical health than mental health of older adults. Mental health is different because when family members and friends share health information, it satisfies older adults’ need for care and attention [52], reduces loneliness [14], and improves their well-being and mental health. In addition, social media use had two additional benefits for mental health of older adults that were not relevant to physical health.

Social media satisfied older adults’ emotional needs. Older adults have a strong need to communicate with their family members and friends [53,54], and adequate emotional support can alleviate their negative emotions and improve their mental health [55]. In the past, people usually communicated face to face, and due to the restrictions of many subjective and objective factors, it has been difficult for older adults to fully meet their needs for social interaction. Social media, however, can overcome the influence of traditional unfavorable factors, enabling older adults to communicate with their family members and friends through diverse means such as texts, pictures, voice, and video. This meets older adults’ emotional needs substantially.

Social media maintains and expands the social networks of older adults. The social network has an important impact on the health of older adults [56,57], and a good social network can enhance their subjective well-being and life satisfaction [35]. Restricted by retirement as well as mobility and health factors, older adults’ contact with their original social network is weakened [58], and a new network is difficult to establish [33]. This easily leads to negative emotions [59]. Social media allows older adults to maintain their original social networks through online interaction. This behavior has mitigated the negative psychological effects of weakening social networks on older adults [36,38,60], and the openness of social media can help them make new friends with common interests, expand social contacts, and establish a new social network [19].

Although this study analyzed the effects of social media use on the health of older adults and analyzed possible mechanisms of action, there are several limitations herein. First, although the mechanism of the influence of social media use on the health of older adults has been analyzed, the results of the study only prove that social media use is significantly related to the health of older adults. Due to the lack of controlled experiments, this study could not determine whether social media use has a significant impact on the health of older adults. However, it is not possible to directly analyze the effects of the mechanism. The mechanism analysis mainly adopts the existing research as its theoretical basis, and the data in this paper mainly verify its final conclusion at the empirical level. Subsequent research can pay more attention to the mechanism through which the use of social media affects the health of the elderly, the different ways in which Internet use affects health, the different ways in which it impacts different people and whether there is an effect of population size.

Second, although this paper chose six variables as control variables from the two dimensions of personal characteristics and social characteristics, it may have ignored some factors that can affect the health of older adults, such as personality characteristics, lifestyle habits, health literacy level, etc. Internet use may not be the only cause of differences in physical or mental health among older adults. Therefore, further research should more fully consider the influence of other factors by building more complex and comprehensive models or by adding other control groups to more accurately measure the net effects of Internet use on the health of older adults. Our future research will address these deficiencies.

Finally, although the 2017CGSS data used in the study were the latest available data, there was no problem regarding the reliability of the data or the suitability of the research questions. However, these were second-hand data from five years ago. With the rapid development of the Internet in China, the impact of Internet use on people is also deepening. The conclusions drawn from data from five years ago may not be applicable today, especially given the impact of COVID-19 on people’s Internet use habits. Further research should track the latest data and fully consider the impact of COVID-19 on people’s Internet use and their health, especially mental health. In future research, we will try to independently obtain primary data in a small area (such as Tianjin) to study the impact of Internet use on the health of older adults in the post-epidemic era. In addition, there was a four-years gap between data collection and analysis, and the study did not analyze the consequences of this. It is one of the limitations of this study.

## 5. Conclusions

First, there was a significant positive correlation between social media use and health of older adults. Second, the correlation between social media use and mental health of older adults was more significant than physical health. Based on the above conclusions, the following suggestions are proposed: It is necessary to actively guide older adults to use the Internet, as well as popularize Internet and smart terminals such as mobile phones and computers. In addition, it is important to guard against the risks of Internet use for older adults.

## Figures and Tables

**Table 1 healthcare-09-01143-t001:** Descriptive statistics of the main variables (*n* = 1278).

Variable	Mean	Std
Explained variable		
Physical health	3.48	1.25
Mental health	3.75	1.01
Explanatory variable		
Social media use	3.74	1.74
Personal characteristics		
Gender	1.50	0.5
Household registration	1.41	0.49
Age	69.25	7.352
Social characteristics		
Social status	4.08	1.78
Family income	51.22	80.17
Family size	2.46	1.61

**Table 2 healthcare-09-01143-t002:** Relationships between social media use and the health of older adults.

Variable	Physical Health		Mental Health			
	Mean	Std	Mean	Std	Number	Percentage (%)
All or almost all	3.58	1.34	3.83	1.12	169	13.22
Most	3.78	1.17	4.02	0.94	259	20.27
About half	3.41	1.13	3.59	1.08	86	6.73
Some	3.52	1.10	3.75	0.95	240	18.78
None or almost none	3.41	1.29	3.69	1.00	270	21.13
I have not used any of these devices	3.14	1.32	3.55	1.03	254	19.87
Total	3.48	1.25	3.75	1.02	1278	100.00

**Table 3 healthcare-09-01143-t003:** Regression results for the effects of social media use on the health of older adults (*n* = 1278).

Variable	Physical Health	Mental Health
	Model 1	Model 2
Social media use	−0.061 ** (0.031)	−0.085 *** (0.031)
Personal characteristics		
Gender = 1	0.386 *** (0.102)	0.355 *** (0.103)
Gender = 2	0 ^b^	0 ^b^
Household registration = 1	0.723 *** (0.112)	0.471 *** (0.114)
Household registration = 2	0 ^b^	0 ^b^
Age	−0.027 *** (0.007)	−0.003 (0.007)
Social characteristics		
Social status	0.204 *** (0.030)	0.215 *** (0.031)
Family income	0.002 *** (0.001)	0.003 *** (0.001)
Family size	0.034 (0.033)	0.020 (0.034)

** *p* < 0.05; *** *p* < 0.01. The numbers in ( ) are the standard deviations. b. Because the argument is redundant, it is set to 0.

## Data Availability

The data presented in this study are available on request from the corresponding authors.

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
