# Peer review of "The Effects of Social Media Use on the Health of Older Adults: An Empirical Analysis Based on 2017 Chinese General Social Survey"

_healthcare, 2021, doi:10.3390/healthcare9091143_

Round 1
Reviewer 1 Report
Very interesting topic and I assume it touches on a relevant aspect of a technology influence. I have some comments that might be helpful to increase the quality of the manuscript.
The research questions need clarity. I think connecting each question with clear wordings to the research gaps would help in increasing understandability.
Page 1, 2nd bottom line error in reference
I think the reason for using secondary data, how the secondary data fit into answering the RQs are missing in the description of section 2.1. Also, it's preferable to have a reference to the data set. It sounds like it’s a public data set, but the background and the context of the data set are missing (or at least I could not understand from the anecdote in 2.1).
“The question A16 was…” what question? Better to give a reference or provide context; otherwise describing the questions without the tag should be fine. The same follows for the rest.
Results are presented in the Methods section; that’s confusing.
What is the takeaway from the results of Table 1?
Page 3, 2nd para “Based on the existing studies, control variables …” what does that mean?
Methods section doesn’t include the data analysis methods; for eg. What are Models 1—4 in the Results section? Why these models? These descriptions are usually expected in the data analysis methods.
I don’t think the evidences are enough to claim that the use of social media has significant influence on physical health, may be the results could conclude there is a significant relation between the two variables, but claiming it as influence is wrong.
The discussion section misses the connection of the results to previous studies, and the originality, significance of outcomes, and the reliability of data as well as generalisability of results. What are the limitations? This may also very well include the consequences of publishing an outcome based on data that’s 5 years old and secondary.
Reviewer 2 Report
This paper presents some interesting findings of the differences in physical and mental health related to internet use. The difference in physical health was something that had not been previously studied in great depth and the authors found some interesting results. The study did control well for demographic factors that could have explained this difference. The following comments are suggestions for how the paper could potentially be improved.
The introduction was clear and concise and summarised the previous research well. The only major thing I thought was missing was some coverage of how internet use influences mental and physical health. This was covered extensively in the discussion section. One or two paragraphs about this literature in the introduction may also help the article.
The method was clear. I wasn't sure why the descriptive results were presented during the methods section. This could have been covered in the results section.
The results were clearly presented and it was interesting as mentioned to see the change in physical health. The change in mental health as the authors mentioned was more expected.
The discussion section is where I think there needs some additions and changes made. There were a number of comments on how the internet can influence mental and physical health which may be true but the results could not necessarily support those claims. As an example, it may be that a personality variable such as extraversion is mediating the relationship between social media use and mental health. It cannot be established from this data that internet use itself was the cause of the difference in mental health or physical health. There may be other factors influencing these relationships which were not measured. Further coverage of the limitations of this study is needed.
Further to this point, it really needs intervention studies to determine whether increased internet use would lead to increased mental and physical health for those whose use is lower. This was not established from this data and we know from health interventions that information itself does not necessarily change behaviour. Again the results are promising but the limitations of these assertions needs to be communicated.
Overall though the results were clearly presented and highlights the need for more research in this area. The paper will benefit from a proof read for spelling and grammar.
Round 2
Reviewer 1 Report
Thank you for the revision. Few suggestions worth considering for improving the quality more:
I suggest refraining from using acronyms in the title.
The last two sentences of the abstract provide the same information. instead, its worth it to mention what could be the consequences (or relevance) of 4 years difference between data collection and analysis. Maybe its a limitation?
The last paragraph of the Introduction section it is better to introduce the data source, explain the acronym.
Instead of "Our specific research ideas were:..." Better to mention that as the scope of the study based on the knowledge gap described above it.
last paragraph section 2.1 research object -> research objects. I will not pick up on language errors, since you are going to have a thorough language check on the whole paper.
What specific points do you want to bring in from Table 1 that is relevant to this study? better to add a couple of sentences describing this.
I think for arriving at conclusions like "The more frequently older adults used social media, the better their physical health" you may need control experiments. Otherwise, it can only say that there is a significant correlation exists between the two variables. The same goes with other similar claims as well, eg, "The more frequently older adults used social media, the better their mental health".
